# EXTending availability of self-management structured EducatioN programmes for people with type 2 Diabetes in low-to-middle income countries (EXTEND)—a feasibility study in Mozambique and Malawi

Emer M Brady [ID],[1] The EXTEND Collaborative, Catherine Bamuya,[2] David Beran [ID],[3] Jorge Correia [ID],[4] Amelia Crampin,[2] Albertino Damasceno,[5] Melanie J Davies,[6] M Hadjiconstantinou [ID],[6] Deirdre Harrington [ID],[6] Kamlesh Khunti [ID],[6] Naomi Levitt,[7] Ana Magaia,[5] Jayna Mistry,[1] Hazel Namadingo,[8] Anne Rodgers,[1] Sally Schreder,[1] Leopoldo Simango,[9] Bernie Stribling,[1] Cheryl Taylor,[1] Ghazala Waheed[10]

For numbered affiliations see end of article.

**Correspondence to**
Emer M Brady; emb29@le.ac.uk

## ABSTRACT

**Background** Globally, there are estimated 425 million people with type 2 diabetes (T2D) with 80% from low-middle income countries (LMIC). Diabetes self-management education (DSME) programmes are a vital and core component of the treatment pathway for T2D. Despite LMIC being disproportionally affected by T2D, there are no DSME available that meet international diabetes federation criterion.

**Methods** The aims were to test the feasibility of delivering a proven effective and cost-effective approach used in a UK population in two urban settings in Malawi and Mozambique by; (1) developing a culturally, contextually and linguistically adapted DSME, the EXTending availability of self-management structured EducatioN programmes for people with type 2 Diabetes in low-to-middle income countries (EXTEND) programme; (2) using a mixed-method approach to evaluate the delivery of training and the EXTEND programme to patients with T2D.

**Results** Twelve healthcare professionals were trained. Ninety-eight participants received the DSME. Retention was high (100% in Mozambique and 94% in Malawi). At 6 months HbA1c (−0.9%), cholesterol (−0.3 mmol/L), blood pressure (−5.9 mm Hg systolic and −6.1 mm Hg diastolic) improved in addition to indicators of well-being (problem areas in diabetes and self-efficacy in diabetes).

**Conclusion** It is feasible to deliver and evaluate the effectiveness of a culturally, contextually and linguistically adapted EXTEND programme in two LMIC. The DSME was acceptable with positive biomedical and psychological outcomes but requires formal testing with cost-effectiveness. Challenges exist in scaling up such an approach in health systems that do not have resources to address the challenge of diabetes.

## STRENGTHS AND LIMITATIONS OF THIS STUDY

⇒ This was not a randomised controlled trial but a feasibility study that included patient and patient-related outcome measures.

⇒ No control group was included in this feasibility study as it was considered unethical at this stage to deny people with type 2 diabetes (T2D) the diabetes self-management education (DSME) programme.

⇒ Standard operating procedures were created and utilised throughout the study for all data capture.

⇒ Data were double entered (and discrepancies corrected with source data) from paper case report forms into a secure web-based database specifically designed for the study.

⇒ Patients with T2D and healthcare professionals involved in delivering their care had direct involvement in the cultural and linguistic adaptation of the UK-based DSME at each location.

## INTRODUCTION

The global estimate of prevalent cases of type 2 diabetes (T2D) is approximately 425 million,[1] and accounts for 10.7% of all-cause mortality in people aged between 20 and 79 years old.[2] There are a further 212 million people thought to be undiagnosed[2] with an overall disproportionate number of cases seen in low-middle income countries (LMIC).[3]

T2D is a progressive chronic condition, when suboptimally managed, can lead to the development of both microvascular and macrovascular complications, including, for example, retinopathy, nephropathy and

neuropathy, heart disease, peripheral vascular disease, resulting in end organ damage in approximately one-third to one half of people with T2D.[4] This is concerning particularly in those health systems, such as in LMICs, that are fragmented, overstretched and under resourced, with intermittent drug supplies and rudimentary clinical training on T2D.

The vast economic burden of T2D includes both direct costs from medical care and indirect costs via loss of productivity or earnings, summing to some estimated $1.3 trillion.[5] It is estimated that the healthcare cost for a person with diabetes is twofold higher than without diabetes. This global public health issue is impacting the world's poorest countries; indeed 80% of all cases of T2D come from LMIC such as those within sub-Saharan Africa (SSA). The current epidemiological transition occurring in SSA, due to rapid urbanisation and nutritional shift, has seen an increase in the burden of T2D.[2] It is predicted that by 2045, 47.1 million people in SSA will have T2D.[6] The additional and increasing strain of T2D placed on their already stretched health systems highlights the need to find cost-effective approaches to reducing the disease burden.

Leading international health organisations promote self-management structured education as the cornerstone of diabetes care, and recommend diabetes self-management education (DSME) programmes as a core component of the treatment pathway for diabetes.[7–9] The overall objectives of DSME are to support informed decision-making, self-care behaviours, problem-solving and active collaboration with the healthcare team and to improve clinical outcomes, health status and quality of life.[10] DSME offer a potential financially viable treatment option for healthcare settings within both high-income countries (HIC) and LMIC.

A recent systematic review highlighted that despite there being a number of studies looking at self-management behaviours in SSA, self-management itself is insufficient in these countries. In particular patients do not engage, or are aware in some cases, of risk reducing behaviours such as physical activity, reducing salt-intake and good foot care.[11] Furthermore, the DSME described in these studies do not meet the standards set-out by the National Institute of Clinical Excellence in the UK, that is, that they include certain components,[9] for example;

► An evidence-base.
► Suits the needs of the person.
► Has specific learning objectives.
► That supports the person in developing attitudes, beliefs, knowledge and skills to self-manage diabetes
► Have a structured curriculum that is theory-driven, evidence-based and resource-effective with supporting materials, and is written down.
► Delivered by trained educators
► Is quality assured.

These principles are supported by the American Diabetes Association, European Association for the Study of Diabetes[7] and the International Diabetes Federation.[8] To the best of our knowledge, there are no DSME programmes with proven effectiveness and cost-effectiveness in SSA that meet these criteria.

The EXTEND programme was a cultural and contextual adaptation of a UK DSME that meets international criteria for DSME and has previously been shown to be effective and cost-effective in people with T2D. The aim of this study was to test the feasibility of the EXTEND programme including; working with local teams to deliver training, recruiting patients, delivering the programme and collecting biomedical and psychological research outcomes in two SSA urban settings in Malawi (Lilongwe) and Mozambique (Maputo).

## METHODS

This was a single group feasibility study with mixed-methods evaluation. All participants received the intervention. This study was funded by the Global Challenges Research Fund NCDs Foundation Awards 2016 Developmental Pathway Funding (Medical Research Council (MR/P02548X/1)). All participants were offered the intervention. Data were collected after informed consent was obtained and before the intervention (baseline) and at 6 months. The detailed Consolidated Standards of Reporting Trials diagram is provided in online supplemental material 1. Briefly in Lilongwe, baseline data were collected in April 2018, DSME delivered in May 2018 and follow-up data in October 2018. In Maputo, baseline data and DSME delivery took place in June 2018 and follow-up in December 2018. The qualitative study was conducted in Lilongwe February 2019 and in August 2019 in Maputo.

### Participants

Patients were recruited from the private diabetes outpatients clinic provided by the Mozambican Diabetes Association (AMODIA), in Maputo. In Malawi, patients were identified from the government-funded health centre in Area 25 in Lilongwe. In both settings, the patients' paper health records were examined for eligibility. Eligibility were a diagnosis of T2D and 18 years old or older. Exclusion criteria were: severe and enduring mental health problems; not primarily responsible for their own care, could not provide informed consent; not able to participate in activities in a group setting or currently participating in another intervention study. Those identified as eligible were either approached at their next appointment or telephoned by one of the study researchers. The study was explained and if they were interested they were invited to attend the baseline assessment visit. All participants had at least 24 hours to consider participation and could withdraw at any time without their usual care being affected.

### Sample size

No formal sample size was calculated because this is a feasibility study. A sample size of 50 participants in each

site was selected based on a balance between pragmatism and having a large enough sample to produce reasonable parameter estimates to power a future formal evaluation of the EXTEND programme and experience of the logistics of its delivery.

## The intervention

The DSME programme, named 'EXTEND', aimed to extend the availability of self-management structured education programme for people with T2D in LMIC. EXTEND is an interactive group-based programme culturally and contextually adapted from a programme first developed and tested in the UK[12–14] and meets international guidelines for DSME. It was delivered in two 3-hour sessions by two trained educators to people with T2D in Portuguese in Maputo and Chicheŵa in Lilongwe. The DSME was delivered within 3 weeks of the baseline data collection visit.

The programme has a written curriculum and educators were trained to elicit the learning of the participants by adopting a non-didactic approach to the group learning. A large part of the curriculum is focused on lifestyle factors, such as food choices, physical activity and cardiovascular risk factors. The UK DSME (DESMOND) and thus the EXTEND programme aimed to activate the participants to explore their own personal risk factors and from this generate achievable goal(s) with an action plan while considering barriers and enablers. The whole programme is underpinned by several learning and behaviour change theories including; the dual process theory, self-efficacy, the social learning theory and Leventhal's common sense theory as described by Skinner *et al*.[15]

## Patient and public involvement

Patient and public involvement (PPI) was of critical importance during the adaptation of the UK-based DSME programme. The EXTEND programme was coproduced by the EXTEND investigators and patients, educators, nurses and patients spouses/children. The UK DSME with supporting resources were taken to Mozambique and Malawi separately by the national educators. At site the local research teams had invited patients and those who would eventually be trained to deliver the education to be part of the PPI group and attend a 2-day session where all content were shared and scrutinised by the local PPI group. Sections of the DSME that were not relevant to the local population were removed or amended. Local foods and cooking practices were included as directed by the PPI groups and other topics of importance, for example, erectile dysfunction and natural remedies for diabetes. The adaptations were made back in the UK in collaboration with the local researchers. The DSME was then translated which prompted further amendments given differences on vernacular. The UK national educators then returned the each locality and delivered EXTEND to two new groups of patients and their spouse/child and received further alterations. During the delivery of EXTEND in the study, further suggestions made during the study were collected and feedback and incorporated into the final version. Please see online supplemental material 2 for key adaptations.

## Educator training

Educators who delivered EXTEND were trained by accredited national educator trainers from the UK who conducted 4-day training sessions in each of the settings; a total of eight educators were successfully trained in Lilongwe and four in Maputo (online supplemental material 3). The trainers' set-up a WhatsApp group with each group of educators to provide ongoing support remotely as requested by the educators. In Malawi, the people trained consisted of three nurses and five lay people with T2D. The research associate at location selected four of the most competent educators to take part in the feasibility study which included two nurses and two lay people with T2D. The remaining four went back into their communities to spread the messages. In Mozambique, the people trained included three nurses and a medical student all four took part in the feasibility study.

Participants and educators were provided refreshments and refunded travel expenses at both baseline, education and follow-up visits.

## Feasibility-related outcomes

Feasibility-related outcomes were collected via a recruitment log completed by the onsite recruitment team:
1. Number of eligible patients referred who accepted the invitation and number who refused
2. Number of eligible patients referred who accepted the feasibility study invitation and attended DSME and research study visits (baseline only, baseline and follow-up)
3. Data collected at each visit
4. Baseline characteristics of the sample who were enrolled in the study
5. Retention rate
6. Change values for each of the potential outcome measures

## Participant outcome data

Predata and postdata were collected from consenting participants at baseline and 6 months. Data were recorded in a paper-based case report form (CRF). Standard operating procedures (SOPs) were developed, agreed and followed at both sites. Data were double entered into a REDCap[16] database, which uses a 'My Structured Query Language' (MySQL) database (an open-source relational database management system (RDBMS)) via a secure web interface, with data checks used during data entry to ensure quality. All supporting systems were hosted and housed within the secure networked environment provided by the University of Leicester, UK.

Data collected are provided in online supplemental material 4 but in summary include demographics, medical history, anthropometric measurements, cardio-metabolic-related

outcomes, psychological outcomes pertaining to health and well-being and lifestyle behaviours.

## Qualitative study

The qualitative study was conducted in both settings with the purpose of exploring the views and experiences of those directly involved with the EXTEND programme (ie, people with T2D, trainers and educators). In addition, the views of clinicians and stakeholders who are regularly in contact with people with T2D were explored on the potential for future implementation. Findings from this qualitative study are presented in detail elsewhere. Briefly, focus group discussions were conducted in the Faculty of Medicine premises (Maputo), and in Area 25 health centre (Lilongwe) (August 2018 to April 2019) with discussions lasted approximately 90 min. The focus groups were carried out by our research team (MH, CB and JC) and audio recorded. MH, who has extensive experience in qualitative research, led data collection and analysis. Where required, research members also acted as translators (JC).

Here, an overview of the participants' views on and experience with the EXTEND programme is provided.

## Analysis
### Quantitative data analysis

Descriptive statistics were produced for participant characteristics at baseline using mean (SD) for normally distributed variables, median (IQR) for non-normally distributed variables, count and percentage (%) for categorical variables. Each of the outcome measures has been summarised using appropriate descriptive statistics at baseline and at 6 months. A paired t-test for normally distributed continuous variables (or Mcnemar's test for categorical variables) was used to compare baseline and post intervention means (or proportions) separately in each country. Non-parametric Wilcoxon ranksum test was used to compare baseline and follow-up medians when the data were not normally distributed.

### Qualitative data analysis

Taking an inductive thematic approach, data were analysed by two researchers (MH and CB) based on the Framework method[17] and applying principles of the constant comparative techniques.[18] Data were organised with the use of the NVivo qualitative data indexing software. An initial coding framework was generated, and further refined through additional coding against transcripts. Data were subsequently summarised and exported into matrices to enable comparison of themes systematically. To ensure credibility, we used investigator triangulation,[19] whereby the two researchers (MH and CB) coded and analysed the data for both localities.

## RESULTS
### Recruitment and retention

A total of 122 were invited to participate across both sites with a total of 12 declining to participate. Reasons for declining included a lack of time, deficiencies for authorisation of the work place, unwillingness to attend on Saturday due to family ceremonies, difficulties in communicating in Portuguese. A total of 98 participants, 50 in Malawi and 48 in Mozambique were recruited to the feasibility study. Overall, the mean age was 55.2 years and 62% were female. The retention rate was high (online supplemental material 1). All outcome data were successfully collected in both sites for both study time-points with the exception of the objective measure of physical activity, which was collected in Malawi at baseline only. In pairs, the educators delivered EXTEND to all participants.

In Lilongwe, all data were collected and the first education session delivered at Malawi Epidemiology and Intervention Research Unit (MEIRU) premises where participants were asked to travel for a distance of about 20 kms. The participants requested the second education session to be delivered at the Area 25 health clinic due to reduced travel time and expense and increased accessibility. Participants preferred all appointments in the afternoon. The data were collected by the research associated assigned to the study with support from a nurse. In Maputo, all data were collected and the education delivered at the AMODIA out patients' clinic situated on the grounds of the Hospital Central de Maputo in the city centre. The data were collected by the two research associates assigned to the study. Data from both sites were recorded in a paper-based CRF. Data were double entered into a REDCap database, which uses a 'My Structured Query Language' (MySQL) database (an open-source relational database management system (RDBMS)) via a secure web interface, with data checks used during data entry to ensure data quality. All supporting systems were hosted and housed within the secure networked environment provided by the University of Leicester, UK.

### Baseline characteristics (table 1)

Twenty-two per cent of participants were within the accepted 'normal' range for BMI, 35% were overweight and 43% as obese. Overall, between 60% and 78% had hypertension with a low prevalence of hypercholesterolemia in Malawi compared with 41% of the Mozambican cohort. A family history of T2D and/or hypertension was common (online supplemental material 5). Glycated haemoglobin (HbA1c) was similar between the two cohorts and indicative of poor glycaemic control. Fasting glucose was higher in Mozambique but in both settings was >7.0 mmol/L, again indicating suboptimal control. Despite total cholesterol levels being within the 'healthy' range, HDLc was low 'normal' and LDLc was high 'normal' which are considered to be associated with increased risk of heart disease. Current medication is provided in online supplemental material 6; no participants were managing their T2D with diet and lifestyle only. The majority were on dual therapy of metformin and sulphonylurea and none on insulin therapy at the time of the study in Malawi. In Mozambique, the majority were using monotherapy (>70%). Over 60% in both

**Table 1** Baseline characteristics

|  | Malawi n=50 | Mozambique n=48 | Overall n=98 |
|---|---|---|---|
| **Demographics** |  |  |  |
| Age | 56.2 (11.6) | 54.2 (7.8) | 55.2 (9.9) |
| Gender (female) | 30 (60.0) | 31 (64.6) | 61 (62.2) |
| Duration T2DM (years) | 6.8 (5.5) | 8.81 (5.9) | 7.79 (5.8) |
| **Medical history (n, %)** |  |  |  |
| Hypertension | 39 (78.0) | 29 (60.4) | 68 (69.4) |
| High cholesterol | 2 (4.0) | 20 (41.7) | 22 (22.5) |
| Last time cholesterol checked (months)* | 0 (0–9) | 1 (1–2) | 1 (0–3) |
| Stroke | 5 (10.0) | 1 (2.1) | 6 (6.1) |
| Heart disease | 0 (0.0) | 1 (2.1) | 1 (1.0) |
| Tuberculosis | 4 (8.0) | 4 (8.3) | 8 (8.2) |
| **Biomedical characteristics** |  |  |  |
| HbA1c (%)* | 9.7 (7.9–14.7) | 9.6 (7.6–14.7) | 9.6 (7.7–14.7) |
| HbA1c (mmol/mol)* | 102.6 (48.4) | 95.1 (46.5) | 98.9 (47.4) |
| Fasting glucose (mmol/L)* | 7.5 (5.5–10.3) | 9.0 (6.5–13.2) | 8.1 (6.2–12.0) |
| Total cholesterol (mmol/L)† | 5.1 (1.3) | 4.7 (0.9) | 4.9 (1.2) |
| HDL (mmol/L)† | 1.3 (0.3) | 1.2 (0.3) | 1.2 (0.3) |
| LDL (mmol/L)† | 3.4 (1.0) | 2.8 (0.9) | 3.1 (1.0) |
| Triglycerides (mmol/L)* | 1.5 (1.0–1.9) | 1.1 (0.8–1.6) | 1.3 (0.9–1.8) |
| Systolic BP (mm Hg)† | 136.8 (20.6) | 142.8 (22.6) | 139.7 (21.7) |
| Diastolic BP (mm Hg)† | 86.8 (8.9) | 89.6 (10.8) | 88.2 (9.9) |
| Weight (kg)† | 71.2 (13.9) | 82.2 (17.8) | 76.5 (16.7) |
| BMI (kg/m$^2$)† | 27.4 (5.3) | 29.5 (6.4) | 28.5 (6.0) |
| **BMI categories** |  |  |  |
| Normal (20–24.9 kg/m$^2$)† | 11 (23.4) | 10 (21.3) | 21 (22.3) |
| Overweight (25–29.9 kg/m$^2$)† | 19 (40.4) | 14 (29.8) | 33 (35.1) |
| Obese (≥30 kg/m$^2$)† | 17 (36.2) | 23 (48.9) | 40 (42.6) |
| Waist circumference (cm)† | 91.8 (12.6) | 95.6 (15.5) | 93.6 (14.2) |

*Median (IQR) for non-normally distributed variables.
†Mean (SD) for normally distributed variables.
BMI, Body Mass Index; BP, blood pressure; HbA1c, glycated haemoglobin; HDL, high density lipoprotein; LDL, low density lipoprotein; T2DM, type 2 diabetes mellitus.

cohorts were taking antihypertensive medication and a quarter of those in Mozambique were on lipid-lowering medication. No participants in Malawi reported being prescribed lipid lowering medication. Data were not collected on medicine adherence. There was a shift from monotherapy (metformin) to dual therapy from baseline to follow-up; metformin plus sulphonylurea increased by 5% and metformin plus insulin by 7%. There was a small reduction in those receiving a diuretic at 6 months.

### Changes in biomedical and psychological outcomes from baseline (tables 2 and 3, respectively)

Although, this feasibility study was not powered to detect statistically significant differences in biomedical outcomes, overall the reductions in HbA1c ($-0.9\%$ (95% CI: $-1.4$ to $-0.1$)), total cholesterol ($-0.3$ mmol/L (95% CI: $-0.4$ to $-0.1$)), low density lipoprotein (LDL) cholesterol ($-0.2$ mmol/L (95% CI: $-0.3$ to $-0.0$)), triglycerides ($-0.2$ mmol/L (95% CI: $-0.3$ to $-0.1$)), diastolic ($-6.1$ mm Hg (95% CI: $-8.2$ to $-4.0$)) and systolic ($-5.9$ mm Hg (95% CI: $-9.6$ to $-2.1$)) blood pressure are clinically important and do reach statistical significance. Heart rate, weight and BMI increased negligibly in both cohorts. A clinically relevant reduction in HbA1c, glucose and diastolic blood pressure are observed in participants in Malawi with modest reductions also observed for total cholesterol, LDL cholesterol and weight at 6 months follow-up. The same pattern is seen in participants from

**Table 2** Mean change in biomedical characteristics from baseline

| Characteristic | Malawi | | | | Mozambique | | | | Overall | | | |
|---|---|---|---|---|---|---|---|---|---|---|---|---|
| | Baseline | Follow-up | Mean change n=47 | P value | Baseline | Follow-up | Mean change n=48 | P value | Baseline | Follow-up | Mean change n=95 | P value |
| HbA1c (%)** | 9.7 (7.9 to 14.7) | 9.8 (7.6 to 13.3) | -0.8 (-1.3 to 0.1) | 0.058 | 9.6 (7.6 to 14.7) | 8.4 (7.2 to 10.7) | -1.1 (-2.5 to 0.1) | 0.144 | 9.6 (7.7 to 14.7) | 8.8 (7.4 to 12.1) | -0.9 (-1.4 to -0.1) | <0.001 |
| HbA1c (mmol/mol) | 102.6 (48.4) | 89.5 (39.5) | 9.7 (32.2) | | 95.1 (46.5) | 77.3 (31.1) | 17.8 (39.3) | | 98.9 (47.4) | 83.4 (35.8) | 13.8 (35.9) | |
| Glucose (mmol/L)** | 7.5 (5.5 to 10.3) | 7.6 (5.6 to 10.7) | -0.4 (-0.9 to 0.4) | 0.589 | 9.0 (6.5 to 13.2) | 7.7 (5.8 to 12.4) | -0.9 (-1.9 to 0.4) | 0.372 | 8.1 (6.2 to 12.0) | 7.6 (5.7 to 12.1) | -0.4 (-1.1 to 0.03) | 0.313 |
| Total cholesterol (mmol/L) | 5.1 (1.3) | 4.9 (1.2) | -0.2 (-0.4 to -0.0) | 0.046 | 4.7 (0.9) | 4.4 (1.0) | -0.3 (-0.6 to -0.1) | 0.016 | 4.9 (1.2) | 4.6 (1.1) | -0.3 (-0.4 to -0.1) | 0.002 |
| HDL (mmol/L) | 1.3 (0.3) | 1.3 (0.3) | -0.1 (-0.1 to 0.0) | 0.069 | 1.2 (0.3) | 1.3 (0.4) | 0.1 (-0.0 to 0.1) | 0.087 | 1.2 (0.3) | 1.3 (0.3) | 0.0 (-0.1 to 0.1) | 0.997 |
| LDL (mmol/L) | 3.4 (1.0) | 3.3 (0.9) | -0.2 (-0.3 to -0.0) | 0.017 | 2.8 (0.9) | 2.6 (0.8) | -0.2 (-0.4 to 0.1) | 0.147 | 3.1 (1.0) | 2.9 (0.9) | -0.2 (-0.3 to -0.0) | 0.013 |
| Triglycerides (mmol/L)** | 1.5 (1 to 1.9) | 1.3 (0.9 to 2.0) | -0.2 (-0.4 to -0.1) | 0.008 | 1.1 (0.8 to 1.6) | 1.0 (0.8 to 1.3) | -0.1 (-0.3 to 0.1) | 0.122 | 1.3 (0.9 to 1.8) | 1.2 (0.8 to 1.7) | -0.2 (-0.3 to -0.1) | 0.002 |
| Systolic BP (mm Hg) | 136.8 (20.6) | 138.5 (19.3) | 0.7 (-3.4 to 4.8) | 0.736 | 142.8 (22.6) | 130.4 (18.9) | -12.3 (-18.2 to -6.5) | <0.001 | 139.7 (21.7) | 134.4 (19.4) | -5.9 (-9.6 to -2.1) | 0.003 |
| Diastolic BP (mm Hg) | 86.8 (8.9) | 80.8 (8.9) | -6.1 (-8.9 to -3.3) | <0.001 | 89.6 (10.8) | 83.5 (9.2) | -6.1 (-9.4 to -2.8) | 0.001 | 88.2 (9.9) | 82.2 (9.1) | -6.1 (-8.2 to -4.0) | <0.001 |
| Heart rate (bpm) | 79.7 (14.5) | 80.9 (12.8) | 1.3 (-2.1 to 4.7) | 0.437 | 74.3 (12.4) | 76.3 (10.6) | 1.9 (-1.5 to 5.4) | 0.269 | 77.1 (13.7) | 78.6 (11.9) | 1.6 (-0.8 to 4.0) | 0.178 |
| Weight (kg) | 71.1 (13.9) | 71.0 (12.4) | 0.1 (-0.9 to 1.1) | 0.809 | 82.2 (17.8) | 82.0 (17.2) | -0.2 (-1.3 to 1.0) | 0.781 | 76.5 (16.7) | 76.5 (15.9) | -0.0 (-0.7 to 0.7) | 0.595 |
| Waist (cm) | 91.8 (12.6) | 91.0 (11.7) | -0.9 (-2.8 to 0.9) | 0.323 | 95.6 (15.5) | 98.0 (14.4) | 2.4 (-0.2 to 4.9) | 0.067 | 93.6 (14.2) | 94.5 (13.5) | 0.8 (-0.8 to 2.4) | 0.345 |
| BMI (kg/m$^2$) | 27.4 (5.4) | 27.5 (4.9) | 0.03 (-0.3 to 0.4) | 0.880 | 29.6 (6.4) | 29.5 (6.2) | -0.1 (-0.5 to 0.3) | 0.747 | 28.5 (6.0) | 28.5 (5.7) | -0.0 (-0.3 to 0.3) | 0.892 |

All values are mean (Standard deviation (SD) for normally distributed variables), unless otherwise stated. Mean change (95% CI) or **Median change [range] for non-normally distributed variables provided.
BMI, Body Mass Index; BP, blood pressure; 95% CI, 95% Confidence interval; HbA1c, glycated haemoglobin; HDL, high density lipoprotein; LDL, low density lipoprotein.

**Table 3** Scores for measures from questionnaire at baseline and follow-up

| Questionnaire score | Malawi baseline n=50 | Malawi follow-up n=47 | P value* | Mozambique baseline n=48 | Mozambique follow-up n=48 | P value* | Overall baseline n=98 | Overall follow-up n=95 | P value* |
|---|---|---|---|---|---|---|---|---|---|
| *Health and Well-being related outcomes (PHQ-9)†* | | | | | | | | | |
| Depressive symptoms severity | | | | | | | | | |
| None-minimal (0–4), n (%) | 27 (54.0) | 33 (70.2) | 0.134 | 24 (50.0) | 29 (60.4) | 0.297 | 51 (52.0) | 62 (65.3) | 0.078 |
| Mild (5–9), n (%) | 16 (32.0) | 11 (23.4) | 0.317 | 19 (39.6) | 15 (31.3) | 0.414 | 35 (35.7) | 26 (27.4) | 0.206 |
| Moderate (10–14), n (%) | 5 (10.0) | 3 (6.4) | 1.000 | 5 (10.4) | 3 (6.3) | 0.480 | 10 (10.2) | 6 (6.3) | 0.527 |
| Moderately severe (15–19), n (%) | 2 (4.0) | 0 (0.0) | 0.157 | 0 (0.0) | 1 (2.1) | 0.317 | 2 (2.0) | 1 (1.1) | 0.564 |
| Severe (20–27), n (%) | 0 (0.0) | 0 (0.0) | – | 0 (0.0) | 0 (0.0) | – | 0 (0.0) | 0 (0.0) | – |
| Proportion with score>10, n (%) | 6 (12.0) | 1 (2.1) | 0.059 | 2 (4.2) | 3 (6.3) | 0.564 | 8 (8.2) | 4 (4.2) | 0.206 |
| Proportion with score≥10, n (%) | 7 (14.0) | 3 (6.0) | 0.103 | 5 (10.4) | 4 (8.3) | 0.706 | 12 (12.2) | 7 (7.1) | 0.166 |
| *Problem Areas in Diabetes (PAID)‡* | | | | | | | | | |
| Raw score, mean (SD) | 11.2 (11.9) | 6.0 (8.3) | 0.012 | 21.1 (11.7) | 11.5 (8.6) | <0.001 | 16.1 (12.7) | 8.8 (8.8) | <0.001 |
| Proportion with raw score≥40, n (%) | 2 (4.0) | 1 (2.0) | 0.564 | 5 (10.4) | 0 (0.0) | 0.025 | 7 (7.1) | 1 (1.0) | 0.034 |
| Percentage score, mean (SD) | 14.0 (14.9) | 7.6 (10.3) | 0.012 | 26.4 (14.6) | 14.3 (10.7) | <0.001 | 20.1 (15.9) | 11.0 (11.0) | <0.001 |
| *WHO (Five) Well-Being Index§* | | | | | | | | | |
| Raw score, mean (SD) | 19.1 (5.2) | 20.0 (5.1) | 0.514 | 17.6 (4.7) | 16.1 (4.1) | 0.049 | 18.4 (5.0) | 18.0 (5.0) | 0.382 |
| Proportion with raw score<13, n (%) poor well being | 6 (12.0) | 4 (8.0) | 0.739 | 7 (14.6) | 10 (20.8) | 0.366 | 13 (13.3) | 14 (14.3) | 0.655 |
| Percentage score, mean (SD) | 76.5 (20.8) | 80.1 (20.5) | 0.514 | 70.6 (18.6) | 64.4 (16.6) | 0.049 | 73.6 (19.9) | 72.2 (20.1) | 0.382 |
| Proportion with percentage score<25, n (%) likely depression | 1 (2.0) | 1 (2.0) | 0.317 | 4 (8.3) | 0 (0.0) | 0.046 | 5 (5.1) | 1 (1.0) | 0.180 |
| *Self-Efficacy for Diabetes (DSEQ)¶*, mean (SD) | 68.7 (16.9) | 79.8 (10.2) | 0.001 | 70.7 (12.9) | 69.1 (10.0) | 0.449 | 69.7 (15.0) | 74.4 (11.4) | 0.027 |
| *Medical Outcomes Study: (SF-20) ** * | | | | | | | | | |
| Functioning | | | | | | | | | |
| Physical functioning | | | | | | | | | |
| Median (IQR) | 91.6 (75–100) | 100 (91.6–100) | 0.545 | 100 (100–100) | 100 (95.8–100) | 0.009 | 100 (83.3–100) | 100 (91.6–100) | 0.282 |
| Role functioning | | | | | | | | | |
| Median (IQR) | 100 (50–100) | 100 (100–100) | 0.084 | 100 (100–100) | 100 (100–100) | 0.183 | 100 (100–100) | 100 (100–100) | 0.030 |
| Social functioning | | | | | | | | | |
| Median (IQR) | 100 (100–100) | 100 (100–100) | 0.813 | 100 (100–100) | 100 (100–100) | 0.542 | 100 (100–100) | 100 (100–100) | 0.793 |

Continued

**Table 3** Continued

| Questionnaire score | Malawi baseline n=50 | Malawi follow-up n=47 | P value* | Mozambique baseline n=48 | Mozambique follow-up n=48 | P value* | Overall baseline n=98 | Overall follow-up n=95 | P value* |
|---|---|---|---|---|---|---|---|---|---|
| **Well-being** | | | | | | | | | |
| Mental health | | | | | | | | | |
| Median (IQR) | 88 (76–96) | 92 (84–100) | 0.069 | 80 (68–88) | 76 (64–84) | 0.681 | 84 (72–88) | 84 (72–92) | 0.344 |
| Health perceptions | | | | | | | | | |
| Median (IQR) | 46.8 (32.2–60) | 65 (41.8–90) | <0.001 | 53.6 (30–67.2) | 48.6 (32.2–61.1) | 0.894 | 46.8 (30–62.2) | 56.8 (36.8–76.8) | 0.005 |
| Pain | | | | | | | | | |
| Median (IQR) | 40 (0–60) | 20 (0–60) | 0.284 | 40 (0–60) | 40 (20–60) | 0.407 | 40 (0–60) | 20 (0–60) | 0.171 |

Mean (SD) expressed for normally distributed variables and median (IQR) for non-normally distributed variables.
*P-value calculated using McNemar's test or paired t-test or non-parametric Wilcoxon ranksum test.
†PHQ-9; responses were for over the last 2 weeks; scale for all responses ranges from 0=not at all to 3=nearly every day; total scores range from 0 to 27 with higher scores indicating poor health status.
‡PAID: Problem Areas in Diabetes; scale for all responses ranges from 0=not a problem to 4=serious problem; total scores range from 0 to 80. The score is multiplied by 1.25 to get the percentage score and a score of>=40 is severe diabetes distress.
§WHO Well-Being Index; responses were for over the last 2 weeks; scale for all responses ranges from 0=at no time to 5=all of the time; total raw scores range from 0 to 25, 0 representing worst possible and 25 representing best possible quality of life. The raw score is multiplied by four to obtain a percentage score ranging from 0 to 100.
¶Self-efficacy for Diabetes (Modified version of DSEQ); scale for all responses ranges from 1=not at all confident to 10=totally confident; total scores range from 9 to 90 with higher scores indicating higher self-efficacy.
***SF-20; physical functioning total raw score=18, role functioning total raw score=6, social functioning total raw score=6, mental health total raw score=30, health perception total raw score=25, pain total raw score=6. All parameters scaled 0–100. A higher score indicates better functioning or well-being. The only exception is pain: a higher score indicated more pain.

Mozambique in addition to a significant reduction in systolic blood pressure.

Overall, the prevalence of 'non-minimal presence of depressive symptoms' (as measured by PHQ-9) at baseline was high at 52% and increased at 6 months. There were no reported cases of severe depressive symptoms in either cohort (score 20–27). Malawi and Mozambique had similar levels of mild and moderate depression at baseline (Malawi 32% and 10% respectively, Mozambique 39.6% and 10.4%, respectively) with a lower prevalence for both categories at 6 months. The Problem Areas in Diabetes (PAID) showed a significant improvement with an overall reduction of approximately seven points at 6 months, that is, a lower score is indicative of fewer problems associated with diabetes. A comparable reduction was observed between the two time-points in each setting (Malawi five points and Mozambique approximately nine points). Overall well-being (WHO (Five)) was high in both settings and remained essentially unchanged in Malawi, but there was a reduction in the proportion of those likely to have depression (score<25) in Mozambique at 6 months. Overall, self-efficacy for diabetes improved at 6 months reaching statistical significance. Results from the 20-Item Short Form Health Survey (SF-20) indicated a high and maintained level of physical, role and social functioning. In addition, a reduction in pain and statistically significant improvement in health perceptions at 6 months was observed.

### Qualitative findings (table 4)

Sixty-six individuals were interviewed. Overall, participants who took part shared that the EXTEND programme had a positive impact on their behaviour, indicating improvements in lifestyle habits, including increasing physical activity and improving food choices. They also reported improvements regarding losing weight and taking medication as advised by their doctor (table 4). Participants on a whole expressed positive experience attending a DSME programme such as EXTEND and emphasised a strong need for such self-management education in their local communities. Due to lack of information and education about the management of T2D, the people with this condition had often been misinformed. They had received inconsistent messages about why they have diabetes, what food they should eat, what types of alcohol they should drink or avoid and so on. In addition to the inconsistent information provided to them, they had also been exposed to confusing and conflicting advice from clinicians highlighting that education in diabetes should not only be available for patients, but for healthcare professionals (HCP) also. A detailed analysis of this qualitative study is reported elsewhere.

### DISCUSSION

This study demonstrates that it is possible and acceptable to deliver the culturally, contextually and linguistically adapted EXTEND programme in an urban setting

**Table 4** Qualitative data from patients who received the DSME

| Behaviour change | Patient perspective |
|---|---|
| Improvements in taking medication | I am a living example when I was first diagnosed with T2D I was prescribed to take metformin in the morning and evening but after learning about my condition especially diet topic I was able to manage my diet and as a result I was asked to take metformin once a day and as of now I am only taking half a tablet in the morning and another half in the evening (P5, non-participant, Malawi) |
| Increasing physical activity | For me, this training program was a privilege, and I've been capitalizing on everything I've learned. Especially in the alignment between eating and physical exercise, because this is where I had many problems (P4, patient, Moz) |
| | I could walk from home going to Msungwi but when I reach there I could feel so much pain as if something bad is happening in my body. But after getting the education I have been walking long distances during the evening and I have seen that its working (P5, female patient, Malawi) |
| | The other part which I also liked most was the advice we get from the clinic such as doing different physical exercises like moving a wheel bar, cultivating the garden, so this is helping our bodies to be strong (P6, male patient, Malawi) |
| Improving food choices | We use to abuse on the oil, tomato, onion, everything. Now I know how to do things moderately. I learned a lot, the course was valuable (P4, patient, Moz) |
| | For me I think the program was good because they taught us how to take care of our bodies and the need to consume food that has less cholesterol (P2, female patient Malawi) |
| | We were just ignorant especially on the issue of diet and this was not helping us, but now after getting the education we are able to take care of ourselves (P2, female patient Malawi) |
| | Mmm, and the foods we were taught to eat it's our locally Malawian foods; they didn't tell us to take foreign food which we could spend thousands of money to buy, no so the examples were relevant (P2, female patient Malawi) |
| Manage stress | We stopped having stress and we accepted that we are T2D patients. You also taught us how we can live long by doing exercise, having good diet and etc. The main thing is to accept and avoid stress (P5, male patient, Malawi) |

Continued

| Table 4 | Continued |
|---|---|
| **Behaviour change** | **Patient perspective** |
| Losing weight | I was weighing over 105 kgs but now I have reduced the weight to 92 (P6, female patient Malawi) |
| | I was weighing 85 kg but am weighing 75 kgs and I feel very light now (P1, male patient Malawi) |
| | I already lost 4 kg. My blood sugar levels have already dropped. The tiredness I used to feel I no longer feel (P1, patient, Moz) |
| DSME, diabetes self-management education. | |

in two LMIC. A study to recruit to and collect data for the purpose of evaluating the biomedical and psychological impact of the EXTEND programme was successfully developed and implemented.

A number of key learnings came from this feasibility study. First, additional health information should be collected from participants including; medicine adherence, access to medicine, engagement with traditional healers and use of traditional medicines, previous education in diabetes and the presence of any communicable disease comorbidities, that is, HIV/AIDS. The incomplete objective physical activity data at both sites means that future collection will require dedicated 'hands-on' technology support for device set-up, initialisation and download. This is particularly important as increased objectively measured physical activity in LMICs will progress the field of physical activity surveillance and intervention development.[20]

It was anticipated that the development of the protocol would take approximately 12 weeks, however it took 26 weeks; therefore, a longer time-frame for the development of these core documents should be built into any future work. Due to the large interest and requests from people with T2D for the DSME programme (after recruitment had ceased) as reported by local research teams, it is preferable that a future effectiveness study focuses on less traditional study designs, for example, step-wedged or wait lists, or build in infrastructure/finance to permit control arm participants to receive the programme at the end of the study at no personal cost to them.

The baseline data indicated people with T2D in the two urban settings have poorer glycaemic control than their UK counterparts.[21] This difference extends to the pooled baseline HbA1c from the USA, Sweden and Thailand reported in a systematic review conducted by Steinsbekk and colleagues in 2012.[22] It is well recognised that health outcomes for patients with T2D are largely dependent on the individual's ability to effectively implement and sustain complex self-management skills into their daily lives.[22 23] Encouragingly, while this feasibility study was not powered to detect a change in outcomes at 6 months both clinically and statistically significant

changes in biomedical and psychological outcomes are observed. The reduction in HbA1c, for example, that is approaching 1% is clinically meaningful for example a 1% reduction in HbA1c reportedly reduces the risk for any end point related to diabetes by 21%.[24] Furthermore, reductions in SBP and diastolic blood pressure (DBP) of ≥2 mm Hg are reported to significantly reduce the incidence of CVD,[25 26] thus the reduction of −5.9 and −6.1 (for SBP and DPB, respectively) can be considered clinically meaningful. These data are in-line with evidence from previously tested DSME programmes in HIC.[22] Our data also indicate that these changes are not driven by weight loss as one might expect but may be instead attributed to the reported lifestyle changes (table 4) and a greater understanding of their condition and thus adherence to both glucose lowering and anti-hypertensive medications.

The improvements observed in self-efficacy and knowledge of diabetes are supported by previously reported DSME[27 28] and is the core for successful self-management. Behaviour change is not determined solely by knowledge and information, it is however, fundamental that the individual understands their condition and is equipped with the appropriate skills and confidence to self-manage. The improved diabetes distress score suggests that attending a self-management programme could have a positive impact on behaviour change and emotional well-being (PAIDS).[29] The data from the qualitative study support the acceptability and need for a DSME such as EXTEND in LMICs, given the lack of available education around T2D and their limited access to support for the management of their diabetes.

The rising prevalence of T2D in LMICs, whose healthcare systems are already under immense pressure with infectious disease and limited resources, highlights the need for low-cost effective interventions. This feasibility study demonstrated short-term benefits of the DSME EXTEND that meets international principles for self-management education. To the best of our knowledge, this is the first DSME meeting international guidelines for DSME in Malawi and Mozambique. A definitive trial that includes multiple settings (urban, rural and remote) and cost outcomes is required to formally evaluate the effectiveness and cost-effectiveness of this DSME, and be powered to examine impact on clinical outcomes, diabetes complications with adequate follow-up to explore the persistence of any changes observed in outcome measures. It should also address the sustainability of such programmes in the settings they are tested with implementation pathways and buy-in from influential stakeholders and national decision-makers from the off-set. It is advisable to employ the 'train the trainer' model and use lay educators which has been shown to be as effective as HCP provision in the UK.[30]

Study limitations include it was only delivered in urban settings thus results are not generalisable to the wider diabetes populations for example, rural and remote dwellers. Furthermore, this study was not powered to look at change in biomedical or psychosocial outcomes

nor did we have a control group therefore it cannot be ruled out that the observed changes were due to chance or a maturation effect. The qualitative study followed a robust process to collect and analyse data, however, we acknowledge limitations around the sample size and diversity of individuals' characteristics therefore may not be generalisable to other patients with T2D from other parts of either country. Study strengths include use of standardised CRFs and SOPs at each site, double entry of all data, the successful recruitment of desired sample and high retention rate. All data were collected at each site for both time-points except objective measure of physical activity. Finally, the significant and clinically relevant reduction in biomedical, psychological parameters and the patient experience demonstrate a need for DSMEs such as EXTEND.

## CONCLUSION

It is feasible to train educators to successfully deliver a fit-for-purpose DSME in urban settings in two LMIC. The positive biomedical and psychosocial outcomes observed warrant the formal evaluation of the effectiveness, cost-effectiveness and sustainability of the EXTEND programme in Malawi and Mozambique.

**Author affiliations**
[1]Leicester Diabetes Centre, University Hospitals of Leicester NHS Trust, Leicester, UK
[2]Malawi Epidemiology and Intervention Research Unit, Lilongwe, Malawi
[3]Division of Tropical and Humanitarian Medicine, Faculty of Medicine, University of Geneva and Geneva University Hospitals, Geneva, Switzerland
[4]Unit of Patient Education, Division of Endocrinology,Diabetology, Nutrition and Patient Education, WHO Collaborating Center, Department of Medicine, University of Geneva and Geneva University Hospitals, Geneva, Switzerland
[5]Faculty of Medicine, Eduardo Mondlane University, Maputo, Mozambique
[6]Diabetes Research Centre, College of Life Sciences, University of Leicester, Leicester, UK
[7]University of Cape Town, Rondebosch, Western Cape, South Africa
[8]Malawi Epidemiology and Intervention Research, Lilongwe, Malawi
[9]Mozambican Diabetes Association (AMODIA), Maputo, Mozambique
[10]Health Sciences, University of Leicester, Leicester, UK

**Acknowledgements** Individual REDCap study databases and ongoing data management were developed and conducted in partnership with the NIHR Leicester Biomedical Research Centre Bioinformatics Hub.

**Contributors** EMB: Conceived the study and participated in the study design and co-ordination of the study. Participated in the design of the protocol, case report form and database. Contributed to data cleaning, analysis and interpretation, the development of intervention. EMB drafted the manuscript and co-ordinated and drafting of revisions of manuscript, figures and tables. CB and AM: participated in the study coordination, study design, acquisition of data and interpretation of results, revisions of manuscript. LS: participated in the study coordination, study design, acquisition of data and interpretation of results, revisions of manuscript. JM: Participated in the co-ordination of the study and was involved in the development of intervention, database design, data cleaning and revisions of manuscript. MJD: Conceived the study and participated in the study design, the development of intervention, data analysis and interpretation and contributed to revisions of manuscript. AD: Conceived the study and participated in the study design, designing of the case report form, data interpretation and revisions of manuscript. SS, BS, CT and AR: Contributed to the development and delivery of intervention, data collection and interpretation and to the revisions of manuscript. AC: Conceived the study and participated in the study design, the development of intervention and Case report form. In addition to the database design, data interpretation and contributed to revisions of manuscript. HN: Participated in writing of the protocol and development of the case report form, development of intervention and revisions of manuscript. DB, JC, MH, DH, KK and NL: Conceived the study and participated in the study design, interpretation of results and revision of manuscript GW: Led the statistical analysis, drafting of data tables and contributed to revisions of the manuscript.

**Funding** This work was funded by the Medical Research Council Global Challenges Research Fund NCDs Foundation Awards 2016 Developmental Pathway (MR/P02548X/1).

**Competing interests** KK reports grants and or personal fees from Amgen, Astrazeneca, Baye, NAPP, Lilly, Merck Sharp and Dohme, Novartis, Novo Nordisk, Roche, Berlin-Chemie AG/Menarini Group, Sanofi-Aventis, Servier, Bohringer Ingelheim and Pfizer outside the submitted work. MJD reports grants from University of Leicester and international development fund during this work. DH reports non-financial support from Novo Nordisk during this work.

**Patient consent for publication** Not required.

**Ethics approval** Ethical approval was granted by the Scientific Commission of the Faculty of Medicine and the Mozambique National Research Ethics Committee CNBS—Comité National de Bioética para Saúde (CIBS FM&HCM/80/2017); the College of Medicine Research Ethics Committee, Malawi (P.10/17/2301) and the University of Leicester College of Life Sciences research ethics committee (ref: 145734, 4 April 2018).

**Provenance and peer review** Not commissioned; externally peer reviewed.

**Data availability statement** No data are available. The authors confirm that the data supporting the findings of this study are available within the article and its supplementary materials.Raw data that support the findings of this study are available from the corresponding author (EMB, MJD) upon reasonable request.

**ORCID iDs**
Emer M Brady http://orcid.org/0000-0002-4715-9145
David Beran http://orcid.org/0000-0001-7229-3920
Jorge Correia http://orcid.org/0000-0002-7020-0695
M Hadjiconstantinou http://orcid.org/0000-0003-2827-0988
Deirdre Harrington http://orcid.org/0000-0003-0278-6812
Kamlesh Khunti http://orcid.org/0000-0003-2343-7099

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
