## [Reviewer comments · BMJ Open]

ARTICLE DETAILS

TITLE (PROVISIONAL)	EXTending availability of self-management structured Education programmes for people with type 2 Diabetes in low-to-middle income countries (EXTEND) – A feasibility study in Mozambique and Malawi
AUTHORS	Brady, Emer M.; Bamuya, Catherine; Beran, David; Correia, Jorge; Crampin, Amelia; Damasceno, Albertino; Davies, Melanie; Hadjiconstantinou, M; Harrington, Deirdre; Khunti, Kamlesh; Levitt, Naomi; Magaia, Ana; Mistry, Jayna; Namadingo, Hazel; Rodgers, Anne; Schreder, Sally; Simango, Leopoldo; Stribling, Bernie; Taylor, Cheryl; Waheed, Ghazala

VERSION 1 – REVIEW

REVIEWER	Dinneen, Sean National University of Ireland, Galway, Department of Medicine
REVIEW RETURNED	21-Feb-2021

GENERAL COMMENTS	Summary The authors describe results from a pilot/feasibility study of the delivery of a diabetes self-management education programme (called EXTEND) to 98 patients with type 2 diabetes in 2 sub-Saharan African (SSA) countries. The DSME programme was adapted from the DESMOND programme developed in Leicester, UK and the adaptation included input from educators and public and patient groups in both SSA countries. Although the study was not powered to examine efficacy the authors report statistically significant reduction in several biomedical outcomes. A limited amount of information is provided on a parallel qualitative study involving focus group interviews with educators and participants. The authors conclude that delivery and evaluation of a DSME programme in SSA countries is feasible. Major comments The authors are to be congratulated for delivering a very ambitious body of work involving the adaptation of an existing DSME programme, training of educators and measurement of biomedical and qualitative outcomes in 2 SSA countries. It would be useful to know if options other than adapting DESMOND for SSA were considered; for example developing a more culturally appropriate DSME programme from scratch in one or both countries. Adaptation of DESMOND imposes a set of parameters (eg, 2x3 hour sessions) which may or may not be acceptable to local practice. It would be interesting (and important) to know how much adaptation was required to convert DESMOND to EXTEND. Clearly some of the food-related modules much have changed to
---

	be locally relevant. The authors should include a Table with the different components of DESMOND alongside the components of the adapted EXTEND programme. This would be of interest to the DSME healthcare community. The major learning from a feasibility study is what needs to be adapted or modified to complete a future definitive RCT. The authors should reflect on this using (perhaps) a framework such as ADePT to guide their assessment; https://doi.org/10.1186/1745-6215-14-353 Minor comments P6. Line 10: poor glycaemic control is (hopefully) not the hallmark of type 2 diabetes
--	--

REVIEWER	Huang, Yen-Ming National Taiwan University College of Medicine
REVIEW RETURNED	27-Feb-2021

GENERAL COMMENTS	Abstract  Line 15: ... affected by T2DN there are no DSME available that meet international... → ... affected by T2D, there are no DSME available that meet international Line 24: ... linguistically adapted DSME, The EXTEND programme; 2) evaluating... → ... linguistically adapted DSME, the EXTEND programme; 2) evaluating... Line 26: Data collected at 0 and 6 months. → Data were collected at 0 and 6 months. You need to describe the variables measured in this study as well as the analyses performed in this study. There was no control group in this study. How did you make sure the improvement resulted from the DSME but not maturation? Strengths and limitations of this study  What were the strengths derived from your study? Introduction  P.6 line 6: The global estimates of prevalent cases of Type 2 Diabetes (T2D) are... → The global estimates of prevalent cases of type 2 diabetes (T2D) are... P.6 line 8: ... with a higher proportion seen in Low-middle... → ... with a higher proportion seen in low-middle... P.6 line 37: ... healthcare costs for a person with diabetes is two-fold higher than... → ... healthcare cost for a person with diabetes is two-fold higher than... P.7 line 8: DSME offer a potential financially viable treatment option for... → The DSME offers a potential financially viable treatment option for... P.7 line 21-35: The information in this long sentence is confusing. Please revise it in a clear manner.
---

6. P.7 line 35: These principals are supported by the American Diabetes...
→ These principles are supported by the American Diabetes...
 7. Based on the information in the introduction, it is said that there are no DSME programmes with proven effectiveness and cost-effectiveness in SSA. However, you test the cost-effectiveness and effectiveness of DSME in a UK population in SSA settings. Why did you choose UK population but not local population in SSA?
 8. You need to do a thorough literature review to address why this research question is important. Several studies have proved the cost-effectiveness and effectiveness in a UK population. Why is there a need to repeat the same process to answer the same research question? Is there any rationale that you think the findings from prior research could not be applied to your study? Cultural issues? Sampling issue? Healthcare system issue? What were the findings that have been explored to the effectiveness of the DSME in SSA? What is the existing gap of the effectiveness of the DSME in SSA? Providing this information may shed light on the importance of this study.
- (1) Siegel, K. R., Ali, M. K., Zhou, X., Ng, B. P., Jawanda, S., Proia, K., ... & Zhang, P. (2020). Cost-effectiveness of interventions to manage diabetes: has the evidence changed since 2008?. *Diabetes Care*, 43(7), 1557-1592.
 - (2) Lian, J. X., McGhee, S. M., Chau, J., Wong, C. K., Lam, C. L., & Wong, W. C. (2017). Systematic review on the cost-effectiveness of self-management education programme for type 2 diabetes mellitus. *Diabetes research and clinical practice*, 127, 21-34.
 - (3) Nazar, C. M. J., Bojerenu, M. M., Safdar, M., & Marwat, J. (2016). Effectiveness of diabetes education and awareness of diabetes mellitus in combating diabetes in the United Kigdom; a literature review. *Journal of Nephro pharmacology*, 5(2), 110.

Methods

1. What theoretical framework did you use to inform the development of the study design and the selection of variables in this study?
2. How did you identify the potential participants and perform the process of informed consent?
3. What theory did you use to inform the development of the contents of the DSME program? What was the timeframe of the program and participant recruitment?
4. What kind of study design did you use in this study?
5. What were the variables that were measured in this study? What was your rationale to measure these variables?
6. What were the effect size and power of the quantitative phase in this study?
7. The process of qualitative study was vague, please follow the COREQ checklist to report qualitative study.
8. What were the outcome variables that were measured and analyzed in the quantitative phase?
9. How did you collect and analyze qualitative data? What theory or methodology did you use in the qualitative phase? What kind of coding technique did you used for qualitative data

	analyses? How did you make sure the rigor, trustworthiness, and credibility? 10. How did you measure feasibility of this program? What did you measure to prove the cost-effectiveness and effectiveness in this study? 11. You need to describe the constructs, reliability and validity of the instruments used in this study. Discussion  1. How was the DSME program tailored culturally and contextually, and linguistically? 2. Most of the contents in the discussion were self-evident without the support by the study findings. For example, the improvement of the outcomes may be due to maturation effect but not from the DSME program. How did you prove the impact of the DSME on the participants' outcomes without a control group? 3. There were ample limitations of study design that should be mentioned, such as the threats of validity (e.g., sampling, selection, instrumentation etc.) Please provide more information in this section. 4. What is the uniqueness generated from your study? The findings from your study are the same as previous research about the effects of the DSME program. You need to compare your study findings with existing literature. What are your suggestions regarding implementations for clinical practice and future study? You have to discuss more information relevant to your research question and provide concrete suggestions to enrich existing knowledge of patient care. Reference  1. Please check the reference style is aligned with the journal guideline. For example, the authors and title of references 2, 8, and 12 are cited inappropriately. In addition, you don't need to indicate the date of published online first.
--	---

REVIEWER	Curtis, Ffion University of Lincoln, Lincoln Institute for Health
REVIEW RETURNED	08-Mar-2021

GENERAL COMMENTS	This paper reports some important research that has the potential to significantly improve diabetes self-management support in LMICs. Whilst overall the reporting of this study is done well, I would recommend you make use of published reporting guidelines to ensure more complete reporting throughout (also if the trial/protocol was registered online please include reference to this). Apart from this I have made some relatively minor suggestions for your consideration below). You have excellent PPI and some useful recommendation for future studies in this area discussed, this is very interesting research that I look forward to seeing published. Abstract: Remove globally from first sentence as it's there twice. Abstract methods: revise aim/objective 2) as meaning is unclear
---

	Clarify study design (mixed methods single arm pre & post? You could then remove data collected sentence if needed for word count) Strengths and Limitations of this study Items: 4) should it be 'data were' instead of 'data we' 5) location or site may be better than locality Introduction Page 7 of 30/line 21, Sentence beginning 'furthermore' is very long, consider revising Aim: I would re-word this. You could introduce/describe EXTEND. Then aim could be more concise stating to test the feasibility of delivering EXTEND in (if you wish to have more detail I would develop sub aims or specific objectives here also) Methods Include study design here (see abstract comment) Include date for when study was conducted in text Could you include ethics committee reference For a feasibility it would be useful to provide more detail about recruitment, how were they approached, were they given participant information sheets and the opportunity to ask questions, informed of their right to withdraw at any time etc. Also some context with regards to venue for data collection and education, who collected data (useful for future work in this area) The intervention: would it be possible to name DESMOND here with mention of underpinning theory Participant outcome data page 8, line 49: I would revise first sentence to, 'pre and post data were collected from consenting participants at baseline and 6 months' Analysis page 11 of 30/line 47: should outcome measure be plural (measures). Line 56: data plural (data were not) Just some considerations... Whilst the reporting here is done well it is good practice and helpful to adhere to relevant reporting guidelines (http://www.equator-network.org/ CONSORT extension to pilot and feasibility may be appropriate) Did you have a priori criteria for feasibility progression/success as it would be useful to define these here Results Recruitment and retention: as this is identified as one of the feasibility outcomes would it be useful to provide more detail about number of patients referred who accepted/refused to take part in the study etc. Also what was the reason for missing physical activity data. What was the timeline with regards to baseline/education/6 months- did all participants begin education immediately post baseline? Would it be useful to include cut off values for normal (BMI), poor glycaemic control etc here? Qualitative findings: I would remove significant in this context. Can you clarify this sentence 'Due to lack of information and education about the management of T2D, people with this condition are often misinformed'. Where you say people with this condition are you referring specifically to study population or more broadly SSA/Maputo/Lilongwe. It should just be results of study here.
--	---

	Page 20 of 30 where it states ‘also a detailed analysis of this qualitative study is reported elsewhere’, is this unpublished or can you provide reference details If possible it would be interesting to see some examples of the program adaptations made before, and suggested during the program by participants Discussion Page 21 of 30, line 57; can you provide reference for the sentence ‘Objective measures of physical activity is important in LMIC and NCD populations because there is a dearth of evidence in this area’ Page 22 of 30, line 3; you list additional important data such as duration of diabetes for inclusion in future studies, but you have already included/reported this results. I suggest you check and amend this (and the other points on list). Page 22 of 30, line 11: replace circa with approximately Page 22 of 30, line 27: For clarity revise sentence beginning ‘the baseline data indicated...’ I would suggest breaking it up comparing with uk counterparts in one sentence and systematic review in the other, or just list countries with references instead describing review?
--	--

VERSION 1 – AUTHOR RESPONSE

Reviewer: 1

Dr. Sean Dinneen, National University of Ireland, Galway Comments to the Author:

Summary

The authors describe results from a pilot/feasibility study of the delivery of a diabetes self-management education programme (called EXTEND) to 98 patients with type 2 diabetes in 2 sub-Saharan African (SSA) countries. The DSME programme was adapted from the DESMOND programme developed in Leicester, UK and the adaptation included input from educators and public and patient groups in both SSA countries. Although the study was not powered to examine efficacy the authors report statistically significant reduction in several biomedical outcomes. A limited amount of information is provided on a parallel qualitative study involving focus group interviews with educators and participants. The authors conclude that delivery and evaluation of a DSME programme in SSA countries is feasible.

Major comments

The authors are to be congratulated for delivering a very ambitious body of work involving the adaptation of an existing DSME programme, training of educators and measurement of biomedical and qualitative outcomes in 2 SSA countries. It would be useful to know if options other than adapting DESMOND for SSA were considered; for example developing a more culturally appropriate DSME programme from scratch in one or both countries. Adaptation of DESMOND imposes a set of parameters (eg, 2x3 hour sessions) which may or may not be acceptable to local practice.

Thank you for your positive comments. The reviewer raises an interesting point regarding the consideration of options outside of DESMOND and we agree that using DESMOND as a core did bring about its limitations. These were namely, as suggested, to do with session delivery and feedback from the participants highlighted a need for shorter sessions over a longer period of time. At the time of programme development during the funding application stage all collaborators agreed that taking a programme with proven effectiveness and cost-effectiveness was the best starting point. The DESMOND philosophy and its content e.g., the patient story, risk factors, the activity continuum were areas we wanted to take forward to ensure that patients were empowered and gained the skills for successful self-management. The team had existing skills and experience in adapting this program for other chronic conditions such as polycystic ovary syndrome and indeed in its cultural adaptation for the South Asian community in the UK and other populations such as Australia and Qatar. There was a clear strategy with a successful track record for the process of a robust cultural and linguistic adaptation

in each setting that was designed to fit within the funding period and envelope. Part of the feasibility was to determine which components of the EXTEND programme didn't work well or would support successful implementation which will be used in future work.

It would be interesting (and important) to know how much adaptation was required to convert DESMOND to EXTEND. Clearly some of the food-related modules much have changed to be locally relevant. The authors should include a Table with the different components of DESMOND alongside the components of the adapted EXTEND programme. This would be of interest to the DSME healthcare community.

In response to the reviewer's comment, we have included a table that captures the key adaptations that were made during the process of creating EXTEND. We also include the DESMOND module topics and include the reference to the original pilot work of DESMOND. We have also included the brief report from the national trainers of training visit conducted in Malawi in supplementary material 5. The same itinerary was followed in Maputo.

The major learning from a feasibility study is what needs to be adapted or modified to complete a future definitive RCT. The authors should reflect on this using (perhaps) a framework such as ADePT to guide their assessment;

<https://eur03.safelinks.protection.outlook.com/?url=https%3A%2F%2Fdoi.org%2F10.1186%2F1745-6215-14-353&data=04%7C01%7Cemb29%40leicester.ac.uk%7Ca7db8880eec04f2bcd0108d913d7606d%7Caebecd6a31d44b0195ce8274afe853d9%7C0%7C0%7C637562638029558850%7CUnknown%7CTWFpbGZsb3d8eyJWljiMC4wLjAwMDAiLCJQIjoiV2luMzliLCJBTiI6IjEhaWwiLCJXVCi6Mn0%3D%7C1000&data=DzjwFmrBHiCQSSZZg8kyCakYjLvYn9MzF%2BwJTBhacF8%3D&reserved=0>

Thank you for highlighting this important tool. In the working-up of a future definitive RCT we will apply the ADePT decision tool.

Minor comments

P6. Line 10: poor glycaemic control is (hopefully) not the hallmark of type 2 diabetes

We agree this was an inaccurate and poorly worded statement. This paragraph has been edited;

"The global estimate of prevalent cases of Type 2 Diabetes (T2D) is approximately 500 million ¹, and accounts for 10.7% of all-cause mortality in people aged between 20-79 years old². There are a further 212 million people thought to be undiagnosed² with an overall disproportionate number of cases seen in Low-middle income-countries (LMIC)³."

Reviewer: 2

Mr. Yen-ming Huang, UW-Madison

Comments to the Author:

Abstract

- Line 15: ... affected by T2DM there are no DSME available that meet international...**
→ ... affected by T2D, there are no DSME available that meet international
- Line 24: ... linguistically adapted DSME, The EXTEND programme; 2) evaluating...**
→ ... linguistically adapted DSME, the EXTEND programme; 2) evaluating...
- Line 26: Data collected at 0 and 6 months.**
→ Data were collected at 0 and 6 months.

Thank you these items have been addressed in the abstract.

- You need to describe the variables measured in this study as well as the analyses performed in this study.**

This would be preferable however, the word limit of the abstract doesn't permit this added information. However, these data are well described within the main body of the manuscript and supplemental material.

5. There was no control group in this study. How did you make sure the improvement resulted from the DSME but not maturation?

We designed a single arm feasibility study not only because we wanted to offer the intervention to all but its primary purpose was to determine if the EXTEND programme was feasible. We wanted to determine if it was feasible to collect the biomedical and psychosocial outcomes before and after the intervention and were not powered to detect any differences. The data collected would be used to power a future trial. However, because we do not have a control group we cannot definitely say that the improvements that were observed were an intervention effect. We agree that this is a limitation and as such it is included in the strengths and limitations section on page 3 and within this section in the discussion.

Strengths and limitations of this study

1. What were the strengths derived from your study?

Please see strengths and limitations listed on page 3. Further, the last paragraph on page 22 and into page 23.

Introduction

1. P.6 line 6: The global estimates of prevalent cases of Type 2 Diabetes (T2D) are...

→ **The global estimates of prevalent cases of type 2 diabetes (T2D) are...**

2. P.6 line 8: ... with a higher proportion seen in Low-middle...

→ **... with a higher proportion seen in low-middle...**

Thank you this paragraph has been reworked;

"The global estimate of prevalent cases of type 2 Diabetes (T2D) is approximately 500 million ¹, and accounts for 10.7% of all-cause mortality in people aged between 20-79 years old². There are a further 212 million people thought to be undiagnosed² with an overall disproportionate number of cases seen in low-middle income-countries (LMIC)³."

3. P.6 line 37: ... healthcare costs for a person with diabetes is two-fold higher than...

→ **... healthcare cost for a person with diabetes is two-fold higher than...**

4. P.7 line 8: DSME offer a potential financially viable treatment option for...

→ **The DSME offers a potential financially viable treatment option for...**

Thank you these edits have been made.

5. P.7 line 21-35: The information in this long sentence is confusing. Please revise it in a clear manner.

Thank you this has been rewritten;

"The EXTEND programme was a cultural and contextual adaptation of a UK DSME that meets international criteria for DSME and has previously been shown to be effective and cost-effective in people with T2D. The aim of this study was to test the feasibility of the EXTEND programme including; working with local teams to deliver training, recruiting patients, delivering the programme and collecting biomedical and psychological research outcomes in two SSA urban settings in Malawi (Lilongwe) and Mozambique (Maputo)."

6. P.7 line 35: These principals are supported by the American Diabetes...

→ **These principles are supported by the American Diabetes...**

This spelling error has been corrected.

7. Based on the information in the introduction, it is said that there are no DSME programmes with proven effectiveness and cost-effectiveness in SSA. However, you test the

cost-effectiveness and effectiveness of DSME in a UK population in SSA settings. Why did you choose UK population but not local population in SSA?

The UK DSME was developed and tested some 14 years ago by the Leicester Diabetes Centre as part of a large and pioneering body of work. This programme continues to be delivered in England today and indeed in other countries such as Australia and Qatar following a process of cultural adaptation led by the DESMOND collaborative. Developing such complex interventions from scratch is both costly and time consuming. T2D is a chronic condition with complex pathophysiology and effective self-management is a key component for achieving good patient outcomes whether in the UK, Australia or indeed a country in SSA. We have a proven track record of successful adaptation of this programme. The DESMOND philosophy and its content e.g., the patient story, risk factors, the activity continuum are key components that we wanted to take forward to ensure that patients were empowered and gained the necessary skills for successful self-management.

We emphasise that the UK DSME had proven effectiveness and has been previously shown to be cost-effective to highlight the high quality of the starting point. The original work on the UK DSME was separate and predates the current study. We did not consider it necessary to start from scratch because this UK programme exists and ultimately we want the patients in SSA to achieve the same outcomes through effective self-management as those reported in the UK where possible. The EXTEND programme now needs to be tested for its effectiveness and cost-effectiveness in a definitive RCT.

doi: <https://doi.org/10.1136/bmj.39474.922025.BE>

doi: <https://doi.org/10.1136/bmj.e2333>

doi: [10.1136/bmj.c4093](https://doi.org/10.1136/bmj.c4093).

8. **You need to do a thorough literature review to address why this research question is important. Several studies have proved the cost-effectiveness and effectiveness in a UK population. Why is there a need to repeat the same process to answer the same research questions there any rationale that you think the findings from prior research could not be applied to your study? Cultural issues? Sampling issue? Healthcare system issue? What were the findings that have been explored to the effectiveness of the DSME in SSA? What is the existing gap of the effectiveness of the DSME in SSA? Providing this information may shed light on the importance of this study.**

(1) Siegel, K. R., Ali, M. K., Zhou, X., Ng, B. P., Jawanda, S., Proia, K., ... & Zhang, P. (2020). Cost-effectiveness of interventions to manage diabetes: has the evidence changed since 2008?. *Diabetes Care*, 43(7), 1557-1592.

(2) Lian, J. X., McGhee, S. M., Chau, J., Wong, C. K., Lam, C. L., & Wong, W. C. (2017). Systematic review on the cost-effectiveness of self-management education programme for type 2 diabetes mellitus. *Diabetes research and clinical practice*, 127, 21-34.

(3) Nazar, C. M. J., Bojerenu, M. M., Safdar, M., & Marwat, J. (2016). Effectiveness of diabetes education and awareness of diabetes mellitus in combating diabetes in the United Kingdom; a literature review. *Journal of Nephro pharmacology*, 5(2), 110.

Thank you for your comment and we apologise for the lack of clarity. We believe that we are familiar with the evidence in this area. The research question was whether it was feasible to deliver such a programme in LMIC settings particularly with regards to working with local teams to deliver training, recruiting patients, delivering the programme and collecting biomedical and psychological research outcomes. This question has not been answered because such DSME programmes that meet international guidelines do not currently exist in SSA. Furthermore, no cost-effectiveness studies have been conducted in SSA for such programmes. To note Mash et al did demonstrate that DSME is cost-effective in South Africa however, this education programme did not meet international guidelines.

Determining the cost-effectiveness of the EXTEND DSME was not the aim of this particular work. We believe that this information is covered in paragraphs 4 -6 in the introduction.

doi: [10.1016/j.pec.2015.01.005](https://doi.org/10.1016/j.pec.2015.01.005)

Methods

1. What theoretical framework did you use to inform the development of the study design and the selection of variables in this study?

The feasibility study was largely a quantitative piece of work with a qualitative sub-study. No framework was used to design the feasibility it was designed by a medical statistician at the grant application stage. The variables that were selected were demographic, clinical, bio-anthropometric characteristics and psychological wellbeing. They are described in detail in supplemental material 1. These were selected to a) describe that cohort characteristics at baseline, and b) outcomes we predict the programme to have impact on. The local principal investigators led the development of the case report form including specific items that had been included in studies conducted previously in Malawi and Mozambique. This included the WHO world health survey and the WHO STEPs survey conducted in 2009 in Malawi.

2. How did you identify the potential participants and perform the process of informed consent?

Potential participants were identified from their paper medical records. They had at least 24 hours to consider participation. On the day of baseline data collection, group informed consent was taken. The study research associate verbally explained the patient information leaflet, provided time and encouraged any questions from the patient group. Then with each participant one-to-one went through each item on the consent form and the participant initialled each statement they were agreeable with and then signed the form. The study researcher then counter signed the form. This consent for was kept within the participants study pack in a locked filing cabinet at either MEIRU or AMODIA. A summary has been added to page 7.

3. What theory did you use to inform the development of the contents of the DSME program

Thank you we agree that this is important content to include. We have added the following paragraph and references for the development of the UK DSME

“The DSME has a written curriculum and educators were trained to elicit the learning of the participants by adopting a non-didactic approach to the group education. A large part of the curriculum is focused on lifestyle factors, such as food choices, physical activity, and cardiovascular risk factors. The UK DSME and thus the EXTEND programme aimed to activate the participants to explore their own personal risk factors and from this generate achievable goal(s) with an action plan whilst considering barriers and enablers. The whole programme is underpinned by several learning and behaviour change theories including; the dual process theory, self-efficacy, the social learning theory and Leventhal’s common sense theory as described by Skinner al.”

4. What was the timeframe of the program and participant recruitment?

The feasibility study was 6 months data with data collection at 0 and 6 months from baseline. All education was received within 3 weeks of the baseline assessment.

5. What kind of study design did you use in this study?

This was a single group feasibility study we have added the sentence below to paragraph 1 on page 7;

“No formal sample size was calculated because this is a feasibility study. This was a single group feasibility study with mixed-methods evaluation. All participants received the intervention.”

6. What were the variables that were measured in this study? What was your rationale to measure these variables?

The data that were collected are described in supplementary material 1. These measures cover variables for demographic, clinical, bio-anthropometric characteristics and psychological wellbeing. The rationale for collecting these data were to a) describe the cohort characteristics at baseline, and b) the outcomes we predict the programme to have impact on.

7. What were the effect size and power of the quantitative phase in this study?

This was a feasibility therefore doesn't require a formal power calculation as described in paragraph 1 page 7. The sample size was selected based on a balance between pragmatism and having a large enough sample to produce reasonable parameter estimates to power a future formal evaluation of the EXTEND programme and experience of the logistics of its delivery.

8. The process of qualitative study was vague, please follow the COREQ checklist to report qualitative study.

Thank you we agree that the qualitative study is not described in a large amount of detail. The main focus of this manuscript was to describe the feasibility study. However, the participant feedback from the qualitative sub-study we felt adds additional context to the work. The sub-study has been written as a separate manuscript and is currently under peer review. The separate qualitative paper followed the COREQ checklist to demonstrate it was conducted robustly and in line with ethical procedures and reported to a high standard. We have also added detail regarding data collection (page 10) in the methods;

"Findings from this qualitative study are presented in detail elsewhere. Briefly, focus group discussions were conducted in the Faculty of Medicine premises (Maputo), and in Area 25 health centre (Lilongwe) (August 2018 to April 2019) with discussions lasted approximately 90 minutes. The focus groups were carried out by our research team (MH, CB, JCC) and audio recorded. MH, who has extensive experience in qualitative research, led data collection and analysis. Where required, research members also acted as translators (JCC).

We have also provided more information on data analysis (page 11) in response to comment 10 below.

9. What were the outcome variables that were measured and analyzed in the quantitative phase?

Please see supplementary material 1 for detailed list of the outcome variables measured. Further, tables 1- 3 in the manuscripts and supplementary material 2 and 3 for the outcomes analysed.

10. How did you collect and analyze qualitative data? What theory or methodology did you use in the qualitative phase? What kind of coding technique did you use for qualitative data analyses? How did you make sure the rigor, trustworthiness, and credibility?

Please see answer 8 above. To add the qualitative sub-study was approved by the local ethics committees and all researchers complied with ICH-GCP. The researchers who conducted the qualitative interviews/focus groups were experienced social scientists and followed the topic guides designed for the sub-study which again had been reviewed and approved by the local ethnics committees.

In addition, please refer to page 11 where we have added the following sentence to the qualitative analysis section;

"An initial coding framework was generated, and further refined through additional coding against transcripts. Data were subsequently summarised and exported into matrices to enable comparison of themes systematically. To ensure credibility, we used investigator triangulation, whereby the two researchers (MH and CB) coded and analysed the data for both localities."

11. How did you measure feasibility of this program? What did you measure to prove the cost-effectiveness and effectiveness in this study?

In the current study we did not include cost-effectiveness because we were only testing the feasibility of the programme and data collection etc. as previously discussed. Similarly, we did not design an effectiveness study the future definitive RCT will be designed to test the effectiveness and cost-effectiveness of this programme. To measure feasibility the following is described on page 9;

“Feasibility related outcomes were collected via a recruitment log completed by the onsite recruitment team:

1. *Number of eligible patients referred who accepted the invitation and number who refused*
2. *Number of eligible patients referred who accepted the feasibility study invitation and attended DSME and research study visits (baseline only, baseline and follow-up)*
3. *Data collected at each visit*
4. *Baseline characteristics of the sample who were enrolled in the study*
5. *Retention rate*
6. *Change values for each of the potential outcome measures”*

12. You need to describe the constructs, reliability and validity of the instruments used in this study.

Please refer to supplemental material 1 complete with references. The questionnaires are validated.

Discussion

1. How was the DSME program tailored culturally and contextually, and linguistically?

Please refer to the section ‘Patient and Public Involvement’ on page 7. To provide added details we have now added a table in the Supplementary material 4 that describes the key adaptations that were made.

2. Most of the contents in the discussion were self-evident without the support by the study findings. For example, the improvement of the outcomes may be due to maturation effect but not from the DSME program. How did you prove the impact of the DSME on the participants’ outcomes without a control group?

It was not our aim to conduct an effectiveness study. The reviewer is correct that the observed changes may not be due to the intervention rather occurred by chance or due to a maturation effect.

We have added the following on page 21 final paragraph;

“Further, this study was not powered to determine change in biomedical or psychosocial outcomes nor did we have a control group therefore it cannot be ruled out that the observed changes were due to chance or a maturation effect.”

3. There were ample limitations of study design that should be mentioned, such as the threats of validity (e.g., sampling, selection, instrumentation etc.) Please provide more information in this section.

We apologise for this oversight we have now included additional limitations pertaining to the qualitative study.

“The qualitative study followed a robust process to collect and analyse data, however, we acknowledge limitations around the sample size and diversity of individuals’ characteristics therefore may not be generalizable to other patients with T2D from other parts of either country.”

4. What is the uniqueness generated from your study? The findings from your study are the same as previous research about the effects of the DSME program. You need to compare your study findings with existing literature. What are your suggestions regarding implementations for clinical practice and future study?

Thank you this is an important point. The novelty of this study is that to our knowledge there are no existing DSME in Malawi or Mozambique that meet international guidelines. This is the first DSME and we report how feasible it is to deliver such a DSME in existing clinical settings. In addition, we report that it is feasible to collect data for the purpose of evaluating the biomedical and psychosocial impact of a DSME. Finally, we report the observed positive impact of this DSME at 6 months on glycaemia, blood pressure and wellbeing.

We state the following on page 23 and have added an additional sentence for clarity;

“This feasibility study demonstrated short-term benefits of the DSME EXTEND that meets international principles for self-management education. To our knowledge this is the first DSME meeting international guidelines for DSME in Malawi and Mozambique. A definitive trial that includes multiple settings (urban, rural and remote) and cost outcomes is required to formally evaluate the effectiveness and cost-effectiveness of this DSME, and be powered to examine impact on clinical outcomes, diabetes complications with adequate follow-up to explore the persistence of any changes observed in outcome measures.”

5. You have to discuss more information relevant to your research question and provide concrete suggestions to enrich existing knowledge of patient care.

This is the first, to our knowledge, DSME meeting international guidelines in a LMIC. We have demonstrated the feasibility of training local people to deliver the education and provide patient feedback, patient biomedical and patient psychosocial outcomes. There is now a call for the effectiveness to be formally tested in a definitive randomised controlled trial. This trial should include cost-effectiveness and explore implementation at a national level. This is the gap in patient care - there is no DSME for these patients. However, unless effectiveness is proven and it is shown to be cost-effective policy makers and commissioners will not implement it.

Reference

1. Please check the reference style is aligned with the journal guideline. For example, the authors and title of references 2, 8, and 12 are cited inappropriately. In addition, you don't need to indicate the date of publication online first.

Thank you for identifying this reference formatting problem. It has now been rectified.

Reviewer: 3

Dr. Ffion Curtis, University of Lincoln

Comments to the Author:

This paper reports some important research that has the potential to significantly improve diabetes self-management support in LMICs. Whilst overall the reporting of this study is done well, I would recommend you make use of published reporting guidelines to ensure more complete reporting throughout (also if the trial/protocol was registered online please include reference to this). Apart from this I have made some relatively minor suggestions for your consideration below). You have excellent PPI and some useful recommendation for future studies in this area discussed, this is very interesting research that I look forward to seeing published.

Abstract:

Remove globally from first sentence as it's there twice.

Abstract methods: revise aim/objective 2) as meaning is unclear Clarify study design (mixed methods single arm pre & post? You could then remove data collected sentence if needed for word count)

Thank you the extra 'globally' has been removed. Aim/objective 2) has been changed to;

“...2) using a mixed-method approach to evaluate the delivery of training and the EXTEND programme to patients with T2D.”

Strengths and Limitations of this study

Items:

4) should it be 'data were' instead of 'data we'

5) location or site may be better than locality

Thank you these have been addressed.

Introduction

Page 7 of 30/line 21, Sentence beginning 'furthermore' is very long, consider revising

We agree with the reviewer this is a very long sentence. We have split this out using bullet points.

"Furthermore, the DSME described in these studies do not meet the standards set-out by the National Institute of Clinical Excellence (NICE) in the UK, that is, that they include certain components⁸ for example;

- an evidence-base*
- suits the needs of the person*
- has specific learning objectives*
- that supports the person in developing attitudes, beliefs, knowledge and skills to self-manage diabetes*
- have a structured curriculum that is theory-driven, evidence-based and resource-effective with supporting materials, and is written down*
- Delivered by trained educators*
- is quality assured"*

Aim: I would re-word this. You could introduce/describe EXTEND. Then aim could be more concise stating to test the feasibility of delivering EXTEND in (if you wish to have more detail I would develop sub aims or specific objectives here also)

The aims have been reworded;

"The EXTEND programme was a cultural and contextual adaptation of a UK DSME that meets international criteria for DSME and has previously been shown to be effective and cost-effective in people with T2D. The aim of this study was to test the feasibility of the EXTEND programme including; working with local teams to deliver training, recruiting patients, delivering the programme and collecting biomedical and psychological research outcomes in two SSA urban settings in Malawi (Lilongwe) and Mozambique (Maputo)."

Methods

Include study design here (see abstract comment)

Include date for when study was conducted in text

Could you include ethics committee reference

Thank you the details requested have now been included in the first paragraph of the methods section. A consort diagram has been included in supplementary material 6.

For a feasibility it would be useful to provide more detail about recruitment, how were they approached, were they given participant information sheets and the opportunity to ask questions, informed of their right to withdraw at any time etc. Also some context with regards to venue for data collection and education, who collected data (useful for future work in this area)

This detail has now been added to pages 7, 11 and 12

The intervention: would it be possible to name DESMOND here with mention of underpinning theory

Thank you we have now added this additional information on page 11 with a reference included of the original pilot work that does into depth about these underpinning theories.

Participant outcome data page 8, line 49: I would revise first sentence to, 'pre and post data were collected from consenting participants at baseline and 6 months'

Thank you this has now been changed.

Analysis page 11 of 30/line 47: should outcome measure be plural (measures). Line 56: data plural (data were not)

Thank you this has now been changed.

Just some considerations...

Whilst the reporting here is done well it is good practice and helpful to adhere to relevant reporting guidelines

(<https://eur03.safelinks.protection.outlook.com/?url=http%3A%2F%2Fwww.equator-network.org%2F&data=04%7C01%7Cemb29%40leicester.ac.uk%7Ca7db8880eec04f2bcd0108d913d7606d%7Caebecd6a31d44b0195ce8274afe853d9%7C0%7C0%7C637562638029558850%7CUnknown%7CTWfpbGZsb3d8eyJWljoiMC4wLjAwMDAiLCJQIjoiV2luMzliLCJBTil6lk1haWwiLCJXVC I6Mn0%3D%7C1000&data=XNIB5y8fcvX1hAUN%2Fn9y4d7y%2BIEmQmEbHEWwyuT0Q9g%3D&reserved=0> **CONSORT extension to pilot and feasibility may be appropriate**)

This was a useful resource thank you for the sign posting. Whilst the CONSORT extension isn't completely appropriate as this was a single-group feasibility we have completed the checklist to ensure we are reporting all key components and entered 'not applicable' for those items that refer to randomisation.

Did you have a priori criteria for feasibility progression/success as it would be useful to define these here

We did not define any priori criteria for progression or success as this was a standalone feasibility study.

Results

Recruitment and retention: as this is identified as one of the feasibility outcomes would it be useful to provide more detail about number of patients referred who accepted/refused to take part in the study etc. Also what was the reason for missing physical activity data?

We have now added a consort diagram in supplementary material 6. The objective measurement of physical activity proved more challenging than anticipated. The technology requirements needed to support the set-up, initialisation and download of the devices was more problematic than anticipated. The support provided via the UK site at one in-person visit and via email and a supporting video did not take into account the myriad of challenges of supporting remotely. We have now stated in the opening of the discussion section (start of page 22) that dedicated 'hands-on' support for the set up and downloading of the devices is needed in a future trial.

What was the timeline with regards to baseline/education/6 months- did all participants begin education immediately post baseline? Would it be useful to include cut off values for normal (BMI), poor glycaemic control etc here?

The education was received within 3 weeks of the baseline data collection. This information has now been included in the CONSORT diagram (supplementary material 6) and in the text of the manuscript (first paragraph of the methods and within the 'intervention' sub-section). We haven't included breakdown of HbA1c thresholds nor BMI categories the range and mean are provided in table 1.

Qualitative findings: I would remove significant in this context.

This has now been removed

Can you clarify this sentence 'Due to lack of information and education about the management of T2D, people with this condition are often misinformed'. Where you say people with this condition are you referring specifically to study population or more broadly SSA/Maputo/Lilongwe. It should just be results of study here.

Thank you we agree and have changed this section to read...

"Due to lack of information and education about the management of T2D, the people with this condition had often been misinformed. They had received inconsistent messages about why they have diabetes, what food they should eat, what types of alcohol they should drink or avoid and so on. In addition to the

inconsistent information provided to them, they had also been exposed to confusing and conflicting advice from clinicians...

Page 20 of 30 where it states ‘also a detailed analysis of this qualitative study is reported elsewhere’, is this unpublished or can you provide reference details If possible it would be interesting to see some examples of the program adaptations made before, and suggested during the program by participants.

The full qualitative paper is currently under review and as such we cannot provide a reference. In response to reviewer 2 we have added detail regarding data collection and analysis (pages 10 -11). We have included the main topics of DESMOND and the key adaptations made to the programme in creating EXTEND in supplementary table 4.

Discussion

Page 21 of 30, line 57; can you provide reference for the sentence ‘Objective measures of physical activity is important in LMIC and NCD populations because there is a dearth of evidence in this area’

Page 22 of 30, line 3; you list additional important data such as duration of diabetes for inclusion in future studies, but you have already included/reported this results. I suggest you check and amend this (and the other points on list).

Thank you for raising this these data were indeed collected in this feasibility study. This has now been amended to reflect the important data we considered should in addition be collected in future studies. We have also added a reference relating to the need for objective physical activity assessments in LMICs and amended the sentence slightly to reflect the call to action from that paper.

“...additional health information should be collected from participants including; medicine adherence, access to medicine, engagement with traditional healers and use of traditional medicines, previous education in diabetes and the presence of any communicable disease comorbidities i.e. HIV/AIDS.”

Page 22 of 30, line 11: replace circa with approximately **Page 22 of 30, line 27:** For clarity revise sentence beginning ‘the baseline data indicated...’I would suggest breaking it up comparing with uk counterparts in one sentence and systematic review in the other, or just list countries with references instead describing review?

Thank you these items have been addressed. The sentence has been revised;

“The baseline data indicated people with T2D in the two urban settings have poorer glycaemic control than their UK counterparts¹⁹. This difference extends to the pooled baseline HbA1c from the USA, Sweden and Thailand reported in a systematic review conducted by Steinsbekk and colleagues in 2012²⁰.”

VERSION 2 – REVIEW

REVIEWER	Huang, Yen-Ming National Taiwan University College of Medicine
REVIEW RETURNED	03-Jun-2021
GENERAL COMMENTS	This version has been improved its clarity drastically.